# Prospective Matched Case-Control Study of Over-Early P100 Wave Latency in Migraine with Aura

**DOI:** 10.3390/biomedicines11112979

**Published:** 2023-11-06

**Authors:** Foziah J. Alshamrani, Mona Hmoud AlSheikh, Noora Almuslim, Hatem Al Azman, Fahad Alkhamis, Saima Nazish, Hind Alnajashi, Abdulla Alsulaiman

**Affiliations:** 1Department of Neurology, College of Medicine, Imam Abdulrahman Bin Faisal University, Dammam 34212, Saudi Arabiasnmohammadali@iau.edu.sa (S.N.);; 2Physiology Department, College of Medicine, Imam Abdulrahman Bin Faisal University, Dammam 34212, Saudi Arabia; msheikh@iau.edu.sa; 3Neurology Department, King Fahad University Hospital, Dammam, Imam Abdulrahman Bin Faisal University, Dammam 34212, Saudi Arabia; 4Neurology Division, Department of Medicine, Faculty of Medicine, King Abdulaziz University, Jeddah 21589, Saudi Arabia; hindnajashi@gmail.com

**Keywords:** migraine, cortical excitability, latency, aura, habituation, visual evoked potential, P100, VEPs, headache, case-control study, visual system, neuroimaging and pathophysiology

## Abstract

A sizable portion of the world’s population suffers from migraines with aura. The purpose of this research is to describe the findings of a case-control study that was carried out to gain a better understanding of how migraine with aura manifests. The research looked at the P100 delay of the visual-evoked potential in both eyes of 92 healthy people and 44 patients who suffered from migraines with visual aura. All of the participants in the study were recruited from King Fahad University Hospital in Saudi Arabia. Both sets of people had the same ancestry and originated from the same location. Patients who suffered from migraines with aura exhibited a significantly shorter P100 delay in both eyes compared to healthy controls (*p* = 0.001), which is evidence that their early visual processing was distinct. In order to arrive at these findings, we compared people who suffer from migraines with aura to people who do not suffer from migraines and used them as subjects. These findings contribute to the ongoing attempts to bring the disease under control and provide vitally significant new information regarding the functioning of headaches with auras. The primary focus of study in the future should be on determining the nature of the connection between issues with early visual processing and headaches with aura.

## 1. Introduction

Twelve percent of the world’s population suffers from migraines. Patients generally suffer nausea, vomiting, and light and sound sensitivity, as well as moderate to severe headaches [1,2,3]. The neurological disorder migraine causes recurring headaches. Although the actual pathophysiological mechanisms that cause migraines are unknown, various theories have been proposed. Headaches are connected to variations in neurotransmitters like serotonin and dopamine, which constrict and expand brain blood vessels. Several research organizations believe that brain inflammation causes pain. Since migraines commonly run in families, genetic predisposition must be considered. One idea links migraines to neurotransmitters like serotonin, dopamine, and noradrenaline. These changes may activate brain pain circuits, causing migraineurs to experience headaches, nausea, and light and sound sensitivity. Migraines can impair a person and be costly personally and socially [4]. Migraines are linked to anatomical changes in pain-processing brain regions [5]. Despite the need for more research, there is mounting evidence that estrogen, particularly in women, contributes to migraines [5]. To draw significant results from these studies, a thorough statistical analysis must account for all confounding factors [5]. Changes in the functional connection patterns between the visual, auditory, olfactory, gustatory, and somatosensory cortices may explain some sensory processing changes [6]. Migraines cause repeated headaches, light and sound sensitivity, nausea, and vomiting. These symptoms are common migraine symptoms, according to research. Studies have linked neurogenic inflammation and neuropeptides to migraines, although their exact cause is unknown.

Sensory neurons in the trigeminal nerve system release neuropeptides such as CGRP and substance P, causing neurogenic inflammation. These neuropeptides cause neurogenic inflammation. These neuropeptides dilate and permeabilize blood vessels, causing inflammation and pain. Migraines are associated with greater brain concentrations of neuropeptides, including CGRP and PACAP. These neuropeptides may activate brain pain circuits, causing migraines [7,8,9,10]. Migraine sufferers may benefit from a new medication that inhibits the CGRP pathway. Monoclonal antibodies against CGRP or its receptor have been demonstrated to prevent migraines in clinical trials. Neurogenic inflammation and neuropeptides are thought to produce migraines, although further research is needed to confirm this and understand their roles. Electrophysiological testing may reveal migraine headache pathomechanisms. Because they monitor brain and nerve electrical activity, these diagnostic methods can detect minute physiological changes in patients. Research has shown that migraine sufferers have problems with their brains’ sensory absorption and cortex excitability [11,12,13]. These data support the idea that brain neurobiological changes may cause migraines. If researchers can comprehend these changes, they may build better migraine medications. Brain rhythmic irregularities are linked to several neurological and neuropsychiatric diseases. Parkinson’s disease patients had greater cortical gamma-band oscillations and fewer basal ganglia beta-band oscillations. Alzheimer’s disease causes theta and alpha brainwave irregularities in the hippocampus and cortex. Gamma-band oscillations in the prefrontal cortex are disrupted in schizophrenia. Abnormal brain rhythms may be a biomarker for detecting and tracking the development of several diseases and revealing their causes, according to several studies. This study examined how visual evoked potentials (VEPs) adapt to normal and how migraine triggers affect them. The study included 43 migraineurs and 43 healthy controls of the same age and gender. The results show that migraineurs habituate less to VEPs than healthy controls. This contrasts with healthy controls [14,15,16]. Additionally, migraine triggers like stress and sleep deprivation reduced visual-evoked potential habituation. This new study provides important migraine biology information and emphasizes the necessity of identifying and treating migraine triggers. A study of migraine patients without aura who received topiramate examined the use of visual evoked potentials (VEPs) as an anatomical and functional biomarker. Visual evoked potential (VEP) suprasaturation is a non-linear or non-monotone increase in VEP amplitude in response to contrast augmentation. The study found that 53% of migraineurs had VEP supersaturation, compared to 7% of controls. The duration of days between VEP recording and migraine episodes was statistically inversely related to the S index, which measured VEP amplitude responsiveness to contrast gain. All subjects demonstrated this. This relationship was powerful.

### Contribution

This paper makes the following contributions:

This study suggests that the visual cortex is more excitable in the period just before an attack, which lends support to the theory that migraines are the result of cyclical central nervous system dysfunction. They also observed that the P100 amplitude was increased by 23% in migraine patients who did not have aura in contrast to the controls, and that it was equally raised in short-term migraine sufferers who did have aura, but that it demonstrated a drastic reduction with time. This was an interesting finding for the researchers.

The mean latency of the P100 wave in the pattern reversal VEP was longer at the occiput. On the other hand, the mean latency of the P100 waves (P1 and P2) in the single flash VEP was much longer. On the other hand, the mean latency of these waves was typical at the vertex. They came to the conclusion that this deterioration was a result of modest neuronal damage that happened inside the visual system because of recurrent transient ischemia that occurred during the aura. This led them to the conclusion that this deterioration was a consequence of the degeneration of the visual system. This discovery could lend credence to the hypothesis that migraines are linked to alterations in the excitability and plasticity of the cortical circuitry. A current study suggests that P100 may contribute to migraines and that a longer P100 may guard against them. More research is needed to confirm these findings and understand how P100 affects migraine risk.

This research found that migraineurs with auras had greater trouble processing visuals during attacks. These issues may be minor or severe. A scintillating scotoma—a tiny blind spot surrounded by bright, zigzagging lines—is a visual halo. After a foggy or fuzzy zone, normal eyesight returns. The procedure continues here. Neuroimaging shows that migraines with auras modify visual brain activity. These visual processing changes may induce aura-related visual abnormalities.

Our study also revealed that there was a statistically significant difference in the amplitude between migraine patients and the control group for both the left and right eyes, with the control group having bigger amplitude values.

## 2. Literature Review

In most cases, migraine symptoms can be divided into four stages: the prodrome, the aura, the headache, and the postdrome [17,18]. A comprehensive approach to diagnosing and treating migraine symptoms is essential at all stages of the condition [19,20,21,22]. This is because it is generally accepted that migraine symptoms negatively impact both a person’s day-to-day existence and their overall quality of life. There are no visual symptoms connected to a migraine without an aura. Aura-accompanied headaches have been linked to aberrant sensory processing in the brain, including thalamocortical dysrhythmia [23,24], according to some research. Inflammatory neuropeptides and neurotransmitters are released into the brain and bloodstream in response to a migraine episode, setting off a cascade of inflammatory reactions. Migraine sufferers may report worsening headaches as a result of this. Migraines and auras may have their origins in these changes [25,26]. They can either make it easier or more stimulating for the brain to process sensory data. Aberrant activity between the thalamus and the cortex, as might occur with thalamocortical dysrhythmia, is one mechanism underlying sensory hypersensitivity [27]. This is a mechanism underlying heightened sensitivities in the senses. When the suppression of incoming information that the cortex needs to process moves from primary to secondary cortices, visual features become perceptually available [28]. According to electrophysiological studies using instruments like the electroencephalograph (EEG) and the magnetoencephalogram (MEG), migraine attacks are associated with changes in brain activity patterns and excitability [29]. These alterations have been linked to decreased beta-wave activity and increased theta- and alpha-wave activity in some parts of the brain. Migraine-related changes in sensory pathways can be analyzed with noninvasive methods like visual evoked potentials (VEPs) and other types of sensory evoked potentials (SEPs) [30,31,32]. Patients suffering from migraines frequently exhibit these kinds of anomalies.

The cause of migraines has been the subject of numerous studies in recent years. According to studies done on animals, female hormones may be the cause of migraines, at least in part [33]. DNA methylation is just one example of the growing body of research suggesting an epigenetic role in migraine pathophysiology [34]. Although estrogen’s precise involvement in the onset of migraines is unclear [35], a careful evaluation of the data suggests that it plays a significant role. These studies highlight the need for thorough data analysis that accounts for the presence of biases and confounding variables in order to obtain an accurate depiction of migraine pathophysiology. Although they have been recorded during migraine attacks, the reliability of electroencephalograms (EEGs) and visual evoked potentials (VEPs) has not been demonstrated. A SCN1A carrier patient with a migraine in 2016 [36] demonstrated visual evoked potentials (VEPs), although this area still needs extensive investigation. The full range examined migraine can be diagnosed with the help of electroencephalography (EEG) and steady-state visual evoked potentials (VEPs). However, short-lived visual evoked potentials (VEPs) cannot be used for this purpose. However, a recent VEP experiment employing four different spatial frequencies looked into two visual circuits that overlap in migraine sufferers. More research into this area could help scientists figure out what sets off migraines, which would lead to more effective treatments. The S index was found to be correlated with allodynia severity (as judged by the Allodynia Symptoms Checklist, or ASC-12), both at the outset of the study and after three months of topiramate treatment. This was observed prior to and during the patient’s administration of topiramate for the study’s specified treatment period. According to the results of this investigation, the S index shows promise as a biomarker for both the migraine cycle and cortical sensitization. Migraines are thought to originate less from an issue with the blood vessels in the head and more from an issue with the brain’s ability to comprehend sensory information. This is a complex procedure that may require a great deal of effort and several years to perfect. One idea proposes that depression affecting all of the cortical layers is the pathophysiological basis for migraines and auras. Current research is promising and may lead to the development of innovative treatments for migraines without auras, even if it is less obvious what role CSD plays in these cases. Despite the fact that the role of CSD in migraines without auras is less well understood, this remains the case. This inherited brain disorder can cause a variety of symptoms, including headaches, phonophobia, photophobia, and nausea. This study analyzed both an individual’s subjective impressions and existing academic frameworks. After high-frequency grafting, an increase in N2 amplitude in migraineurs is consistent with the idea of cortical hyperexcitability and may indicate a sensory deficit [37,38,39,40]. This information supports the hypothesis that mimics exhibit a heightened sensitivity to stimuli. The purpose of the current study is to compare the early visual processing of patients with migraine and visual aura, as measured by the latency to the P100 wave of the VEPs, to that of healthy persons who have been matched with these patients. This will be done to see if people with migraines and visual auras have different patterns of early visual processing than people without migraines and visual auras.

## 3. Materials and Methods

### 3.1. Research Ethics

During the whole of this investigation, the principles outlined in the Declaration of Helsinki were respected, and the regional human research ethics committee gave their approval for the undertaking. After providing an explanation of the objectives of the study as well as the procedures that were going to be implemented, approval from each participant was then obtained. The ethical review board of Imam Abdulrahman bin Faisal University (IAU) approved the research project. Everyone who took part in the study was assured that their participation was fully voluntary and that they could stop the study at any moment with no negative consequences. The research was conducted with a group of people who volunteered to take part. Throughout the whole of the study, the participants’ rights to privacy and confidentiality were vigilantly guarded in every situation.

Studies comparing children and adolescents with and without headaches found no statistically significant differences in latency amplitudes or reaction times [41]. Whatever the severity of the headache, the truth remained that this was the case. This suggests that, in the time between migraine attacks, most people who have migraines also experience changes in the way their brain processes information. This has diagnostic value because it is an endophenotypic indicator of the illness. However, studies are being conducted on these pathways currently [42,43,44,45]. It is likely that thalamocortical dysrhythmia and low serotonergic tone contribute to the underlying neurological processes, which include reduced pre-activation levels of sensory cortices.

This is explained by the ceiling hypothesis [46], which says that when it comes to evoked potentials, cortical responsiveness drops when it hits a ceiling point, which leads to a habituation response. One of the many possibilities put forth to explain habituation is the ceiling theory [46]. Migraine sufferers, who have a lower pre-activation level, would experience a slowed or nonexistent habituation since the ceiling would be reached later than in healthy people. Because of this, the chance to create a habit would be missed or postponed. The impaired or absent habituation to the testing stimuli can account for migraine patients’ reports of larger-than-usual VEP amplitudes. Several studies, however, failed to find the same results [47,48,49,50,51,52]. They attributed their failure to a variety of methodological issues, such as the lack of blinding or variations in migraine morphologies [53,54].

### 3.2. Study Design and Participants

In the current inquiry, a prospective case-control study was carried out at the King Fahad Hospital of the University (KFHU) in Al-Khobar, which is situated in the Kingdom of Saudi Arabia (KSA). The latency to P100 in the VEP was used as a measurement tool in the research to make a visual processing comparison between migraine sufferers with aura and those who did not have migraines. The sample consisted of a cohort of patients diagnosed with migraine aura consisting of 44 individuals and 92 individuals who did not suffer from migraines (controls). To guarantee the reliability and validity of the findings, the selection of participants was carried out using a rigorous screening procedure, following precise inclusion and exclusion criteria. This was done to ensure that the results would be accurate.

### 3.3. Sample Size

The present study comprised a sample size of 136 participants, consisting of 92 individuals classified as healthy controls (comprising 62 females and 30 males) and 44 persons identified as prospective migraine sufferers (comprising 29 females and 15 males). The determination of the sample size was based on the mean P100 latency values observed in two groups: Group-1, consisting of migraine patients with aura, and Group-2, consisting of controls. The sample size calculation took into account a 95% confidence interval (CI), a statistical power of 95%, and a sample ratio of 2 (Group-2 divided by Group-1). The enhancement of the study’s statistical power was planned through the inclusion of a greater number of controls in the investigation. The average (±standard deviation) of P100 latency values in Group-1 and Group-2 were 106.42 (±6.99)^22^ and 110.47 (±3.35)^22^, respectively.

### 3.4. Sample Method

Consecutive sampling.

### 3.5. Selection Criteria

#### 3.5.1. Inclusion Criteria

This study recruited patients from the headache clinic at King Khalid University Hospital in Al-Khobar, Saudi Arabia, who met the diagnostic criteria outlined by the International Headache Society (IHS) for migraine with aura [55]. Patients were recruited for this study if they fulfilled the diagnostic criteria for migraine with aura [55]. An exhaustive examination was performed on the participants, which included taking headache diaries, doing a bedside neurological examination, performing a fundoscopy, determining the participants’ visual acuity, and conducting an analysis of extraocular movements and visual fields. Participants were asked to provide specific information on the number of years they have been coping with a migraine diagnosis, the number of episodes they suffer each month, the average number of hours that each attack lasts, and the number of days since their most recent migraine attack that they have been migraine-free. The following are the criteria that were used to include participants in the study:(1)A patient must be at least 18 years old.(2)The patient must have aura-type migraine, as described by the diagnostic criteria of the International Headache Society [56].(3)Fixed with or without requiring a visual acuity of 6/6 or finer.(4)The absence of a headache attack at the time of testing; for interventional research involving either animals or people, ethical permission must be obtained from the competent authority, and the accompanying ethical approval code must be mentioned; these requirements must be met before testing may take place.

#### 3.5.2. Exclusion Criteria

To guarantee the consistency and reliability of the findings, several factors were disqualified from consideration:-Participants who have any neurological condition other than migraine that has been medically verified.-Any condition that pertains to ophthalmology.

Exclusion criteria for the research were any medical, surgical, or both types of diseases that would make assessment of visual acuity or VEP difficult or impossible to perform.

### 3.6. Instrument and Data Collection Procedure

VEPs were measured in a manner consistent with the protocol by using a checkerboard pattern that was shown on a television screen. The pattern was black and white. The sequence was seen from one hundred centimeters, and it spanned a visual angle that was equal to 15 degrees by 12 degrees. Candidates were given the directive to focus their attention on a single red dot even while the stimulus changed locations at a rate of two per second. Independent recordings of visual evoked potentials (VEPs) were generated for each eye using pattern reversal (PR) stimuli. These VEPs were then analyzed. The candidate’s hand served as the site for the ground electrode, and the OZ (active electrode) and FZ (reference electrode) placements of the 10–20 international system were assigned to the traditional disc EEG electrodes. Stimulation was performed in the integral field, while the impedance of the electrode was always maintained below 5 kiloohms. Two hundred different experiments were run, and an investigation was carried out beyond time to see whether the results could be correctly repeated by other researchers. All these things, including the latencies of P100, N75, and N140, as well as the peak-to-peak amplitude of P100-N140, were measured by us [57].

### 3.7. Bias, Confounders, and Statistical Analysis

It is essential to have a good understanding of the potential for bias and the elements that might cause confusion in investigations of neurological and neuropsychiatric illnesses. Inaccuracies in conclusions that can be linked back to defects in a study’s design, data collection, or analysis are referred to as having “bias”, and the word “bias” is used to characterize such inaccuracies. It is not always feasible to draw conclusions regarding the direction of causation between an independent variable and a dependent variable. This is because of the presence of confounding factors in the data.

In the fields of neurological and neuropsychiatric research, one of the most common sources of bias is selection bias, which occurs when the study sample is not typical of the population being studied. The results might be overgeneralized, or the disease’s frequency could be underestimated. Either of these are conceivable outcomes.

Confounding factors are a kind of variable that might occur throughout the process of neurological and neuropsychiatric research. Some examples of confounding variables are age, gender, socioeconomic background, concurrent medical diseases, and the use of medication. The failure to adjust for these elements may lead to the formation of erroneous linkages or wrong conclusions.

The reduction of bias and confounding requires careful planning and design of the research investigation, appropriate statistical methods, and adequate controls for relevant confounders. Performing sensitivity analyses, using randomization and blinding processes, and taking into account confounding factors are some of the methods that may be used to accomplish this goal.

In general, researchers looking into neurological and neuropsychiatric illnesses should be aware of the risk of bias and confounding variables, and they should use appropriate statistical approaches to ensure that their results are reliable and trustworthy.

Due to the complexity of migraine pathophysiology, it is necessary to use a careful and organized method for data analysis that takes many biases and confounding factors into account in order to fully understand the condition [35]. Research conducted on women has shown a connection between increased estrogen levels and the pathophysiology of migraines [35]. According to the estrogen withdrawal hypothesis [35], a migraine might be induced by a reduction in estrogen levels below 45–50 pg/mL after a lengthy period of priming. A study reveals that women who have a history of migraines may be more vulnerable to changes in estradiol levels [35]. Migraines are typically associated with women’s menstrual cycles, and this study implies that women who have had migraines in the past may be more susceptible to these changes. The current views about the etiology of migraines place a focus on neurological and/or vascular dysfunction. To have a thorough understanding of the role that estrogen plays in the formation of migraines, more research is required, particularly in a variety of patient groups [35]. It is necessary to conduct an exhaustive statistical analysis that considers all potential confounding factors to draw appropriate conclusions from these types of investigations.

The current investigation attempted to reduce the influence of any biases by synchronizing the ages and genders of the case and control groups. There did not seem to be any unclear or inconsistent variables that might have influenced the findings. The information that was gathered was analyzed using IBM SPSS Statistics version 21 (IBM Corp., Armonk, NY, USA), and descriptive statistics were produced as a result. Means and standard deviations were provided for the variables of age, N75, P100, N145, and amplitude in these data. Calculations were carried out to determine the percentages and frequencies for each gender. The Chi-square test of independence was used to research any potential connections that may exist between migraine patients and either their gender or the control group. This test was also utilized to investigate any potential linkages that may exist between migraine sufferers and the control group. To make group comparisons for mean ages, N75, P100, and N145 values, as well as amplitude, a *t*-test for independent samples was used. It was determined, with the use of Pearson’s correlation coefficient (r), to what extent the N75, P100, and N145 amplitude values in the right and left eyes are connected to one another. This was done by comparing the two sets of data. It was found that a significance threshold of *p* < 0.05 was appropriate.

## 4. Results

The current research enlisted a total of 136 volunteers, 92 of whom were assigned to serve as the study’s control group and 44 of whom were assigned to serve as the study’s migraine group. The migraine group had a significantly larger proportion of female participants (65.9%) compared to male participants (34.1%), while the control group had an even gender distribution, with 67.7% of the participants being female and 33.3% of them being male. However, the research found no statistically significant correlation between the two (*p* = 0.8). Migraine sufferers had a mean age of 37.02 (±13.6), whereas the control group had a mean age of 34.10 (±12.3). The average ages of the two groups were comparable (*p* = 0.22; for details, see Table 1); there was no statistically significant difference between them.

The results of the study showed that the mean P100 latency in the right eye for the migraine group was 88.2 ms (±13.5), whereas for the control group, it was 103.8 ms (±10.1). The mean P100 latency in the left eye for the migraine group was 83.7 ms (±7.2), while for the control group, it was 103.3 ms (±8.1). The control category had longer P100 latency in both eyes compared to the migraine with aura category (by *t*-test, *p* < 0.001). The mean N75, N145, and amplitudes were significantly lower in the migraine categories in both right and left eyes (by *t*-test, *p* < 0.05). The patients with migraine aura exhibited significantly (by *t*-test, *p* < 0.001) shortened VEP P100 latencies in comparison with the non-migraineur controls (see Table 2). Figure 1 shows the block diagram for the methodology used in this paper. Figure 2 illustrates the representation of migraine and VEP. Figure 3 reports the age-group-based percentage of migraine cases. Figure 4 and Figure 5 present a comparison of p100 latency between both groups for the right and left eye, respectively.

A logistic regression analysis was conducted to investigate the impact of P100 on the likelihood of migraine occurrence. The results showed that in the left eye, a longer P100 was associated with the control group (OR = 2.5, *p* = 0.002), indicating that the risk of migraine was 2.5 times lower for longer P100. Additionally, in the left eye, longer P100 was also linked to the control category (OR = 1.1, *p* = 0.14), which suggests that longer P100 was associated with a 1.1 times lower risk of migraine (Table 3).

We investigated the correlation between VEP measurements in the left and right eyes of migraine patients and control subjects. The correlation coefficient (r) of measurements between the left and right eyes in both categories. The findings indicate a significant positive correlation of measurements, including P100, N75, N145, and amplitude, between the left and right eyes (*p* < 0.0001) (Table 4).

In the current investigation, an effort was made to minimize the impact of any potential biases by matching the case group and control group in terms of age and gender distribution. It did not appear that there were any variables that were hazy or inconsistent in any way, which could have altered the results. Following the gathering of information and its subsequent analysis with IBM SPSS Statistics version 21 (IBM Corp., Armonk, NY, USA), descriptive statistics were generated as a consequence of the study. In these data, the variables of age, N75, P100, N145, and amplitude were each given a mean value along with their respective standard deviations. The percentages and rates of occurrence for each gender were calculated using formulas. In order to investigate any potential relationships that could exist between migraine sufferers and either their gender or the control group, the Chi-square test of independence was utilized. This test was also used to evaluate any potential correlations that might exist between people who suffer from migraines and the group that served as a control. A *t*-test was carried out on independent samples in order to make group comparisons for mean ages, N75, P100, and N145 values, as well as amplitude. It was determined, with the help of Pearson’s correlation coefficient (r), to what extent the N75, P100, and N145 amplitude values in the right and left eyes are connected to one another. This was done by comparing the values in both sets of eyes. The comparison of the two collections of data served to accomplish this goal. It was discovered that a significance level of *p* < 0.05 was suitable.

## 5. Discussion

The potential relevance of epigenetic modifications in the pathophysiology of migraines is the subject of further inquiry, which is now in progress. By examining epigenetic markers, researchers have a chance of developing a deeper comprehension of the biological mechanisms that are associated with migraines. There is a pressing need for further conceptual and methodological investigation into the factors that bring about migraines. By taking a more all-encompassing approach to data analysis and utilizing up-to-date techniques and technology, researchers may be able to improve their understanding of this complex ailment and develop more effective therapies and approaches for preventing disease. 

In order to determine whether or not there are any differences, the VEPs of people who suffer from migraines and healthy controls were compared head-to-head in this study. In the past, research has been conducted on the amplitude, latency, intramural differences, and arrangement of VEPs; however, this research was not conducted with the purpose of standardizing a procedure that could be used as a therapeutic tool [58,59,60,61]. The P100 delay, which is a measure of VEP, was substantially shorter in the group who had migraines with aura, as shown by our findings, in comparison to the group that served as the control. Both eyes were affected in the same manner. This quality was present across all members of the control group. Contrary to the findings of earlier studies, Mariani et al. [62] found that when 20 migraine sufferers were compared to healthy controls, the migraine group demonstrated a significantly longer P100 latency. This conclusion contradicts the findings of prior studies. The findings of prior investigations are called into question as a result of this new information. On the other hand, Kennard and his colleagues found that people who suffer from migraines have a longer P100 delay and bigger P100 amplitude [63]. The visual evoked potentials (VEPs) of 20 migraine sufferers were recorded and compared to the VEPs of age-matched controls. Polich and his colleagues carried out the investigation. When comparing full-field and half-field reversing checkerboard stimulus presentations, the researchers found no statistically significant differences in either the latency or amplitude of the responses [64]. The researchers suggested these responses are probably due to a change in the excitability of the cortical circuitry that takes place at various phases of migraine. This is due to the widespread depression and depolarization of the cerebral cortex that are characteristic of the aura that accompanies a migraine.

Independent studies demonstrated that the amplitude of the P100 pulse varied over the course of observation. The amplitude of migraine sufferers was observed to increase when measured in the time interval in between episodes but to decrease when evaluated when they were experiencing an attack, according to this study. They discovered that the amplitudes of both the P100 and the N145 waves had grown in the days leading up to the attack. In addition to this, they found out that the frequency of the waves had shifted. Additionally, a strong association was found between the P100 delay and the length of time the patient had been experiencing headaches [65,66,67]. These findings hint at increased excitability of the visual cortex in the period soon before an attack, which suggests that migraine sufferers may experience cyclical malfunction of the central nervous system. In studies that evaluated visually evoked potentials (VEPs), it was discovered that those who suffer from migraines have a slightly lower P100 amplitude as well as decreased habituation of the N100, N75, and N145. These findings were found in VEPs. This was the conclusion that could be derived from the study when compared to the controls who did not experience migraines. On the other hand, subgroup research that compared the amplitudes of the VEP P100 and N145 in migraine patients with and without aura (MA and MO) revealed no indication of significant increases in amplitudes or habituation in either group of patients. As a consequence of this, there is very little to no difference in the VEP scores of migraine sufferers who have aura in comparison to those who do not have aura.

In this particular study, we have taken into consideration the possibility of selection bias due to the fact that the inclusion and exclusion criteria may not be applicable to all of the instances, and the dataset may be more likely to contain other neurological illnesses. This is a result of the fact that the inclusion/exclusion criterion will not be satisfied by each and every case. The research was limited in its ability to investigate all of the possible sources of misunderstanding. Other neurological problems or mental comorbidities, for instance, were not taken into account in this study. It is possible that the presence of these co-morbidities will have an effect on the findings of the investigations. To minimize the impact of selection bias in subsequent research, participant recruitment should take place among a population sample that is more broadly representative. Other neurological illnesses and mental comorbidities are other examples of potential confounding variables that need to be taken into account and controlled for in research.

## 6. Conclusions

Understanding migraine causes may help create more effective therapies and prevention strategies for these painful headaches. For instance, research has indicated that estrogen is crucial in migraine pathophysiology, especially in women, which might be used to develop hormone-regulating medications. This study might improve migraine treatment outcomes. Understanding key biological processes and risk factors and creating personalized treatment plans may help doctors alleviate migraine symptoms and minimize their frequency and intensity. This study may influence public health programs that target modifiable risk factors, including food, lifestyle, environmental exposures, and epigenetic alterations, to reduce migraines. In our investigation, left and right eye measures were positively correlated. These data included amplitude, P100, N75, and N145. These findings suggest that one eye’s visual evoked potentials (VEPs) may anticipate the others. The results imply that a single VEP test may be sufficient in certain cases, which has substantial therapeutic implications. If so, individuals and healthcare systems may benefit from reduced testing burdens and costs. Visual evoked potentials (VEPs) were used to compare the visual processing of migraine sufferers and controls. Migraine’s underlying mechanism or visual abnormalities may explain migraine sufferers’ variations in brain activity or visual system processing. Migraine sufferers and controls may have different visual system brain activity or processing. More study is needed to understand these results and the link between migraines and visual processing. Despite this, the study sheds information on migraine causes and emphasizes the need for greater research on migraines and visual processing.

### Limitations and Future Directions

The current study shows some key constraints. To begin, the size of the sample is rather small, and it is possible that it does not accurately reflect the greater community of people who suffer from migraines. Therefore, future research should try to expand the sample size to improve the strength of the results and their capacity to be generalized. The second limitation of the research is that it only examines individuals who have migraines accompanied by visual auras. It is essential to ascertain whether the results are applicable to other forms of migraine and headache diseases. In addition, future research should include a headache-free control group so that results may be compared to that group. Thirdly, the research only investigates the P100 component of the visual evoked potential (VEP) response. Other components of the VEP wave complex as well as other electrophysiological measures, such as functional magnetic resonance imaging (fMRI) or magnetoencephalography (MEG), should be investigated to provide a more in-depth comprehension of the neural activity that is associated with migraines. Fourthly, the research did not develop a standardized technique for employing VEP as a clinical tool. Moving forward, future studies should strive to establish typical lower limit values and standardize a procedure that may apply to a clinical tool. The last limitation of the study is that it did not investigate the feature-specific approach to individual symptoms using an objective instrument. It is proposed that more research be conducted to investigate different components of the visual circuits in a symptom-based way aimed at diverse visual symptoms that accompany headaches. Because of this, a more in-depth study of the neurological processes behind migraines will be possible, which might ultimately lead to improved diagnostic and therapeutic options.

The sample size may be inadequate for many reasons. If the population is diverse, including subgroups with varied characteristics, a larger sample size is required to guarantee that the findings are representative of the total population. To detect a tiny variation between two groups, a larger sample size is required. To avoid false positives and negatives, a larger sample size is required if the data are highly variable.

A headache-free control group would assist in controlling for confounding factors that might impact research outcomes. If the research solely included migraine sufferers, it would be impossible to determine whether VEP wave changes were caused by the migraine or other variables like medication usage or stress. Researchers might compare the VEP waves of migraine sufferers and non-migraine sufferers by integrating a headache-free control group. This would reveal migraine-specific VEP wave alterations. Other VEP wave components may help explain migraines and their neurological implications. VEPs are complicated signals with several components. The VEP is supposed to represent visual pathway activity in each component. Researchers might comprehend migraine brain pathways by studying additional VEP components. The VEP’s P100 component may indicate primary visual brain activation. Migraines delay P100. Migraines may be linked to alterations in primary visual cortex processing. VEP N100 is supposed to indicate secondary visual cortical activity. Migraines increase N100. Migraines may enhance secondary visual cortex activity. Researchers might comprehend migraine brain pathways by studying additional VEP components. This knowledge might lead to migraine remedies. In conclusion, introducing a headache-free control group and examining additional VEP wave components might help comprehend migraines and their neurological implications.

## Figures and Tables

**Figure 1 biomedicines-11-02979-f001:**
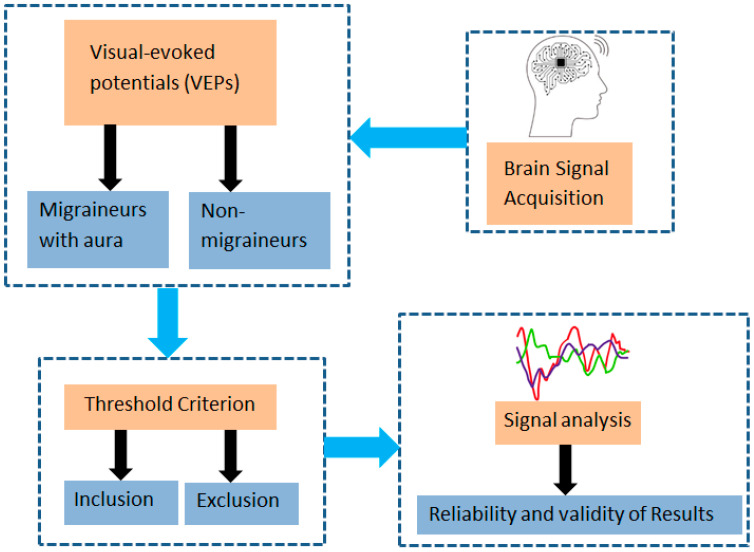
Block diagram for the methodology used in this paper.

**Figure 2 biomedicines-11-02979-f002:**
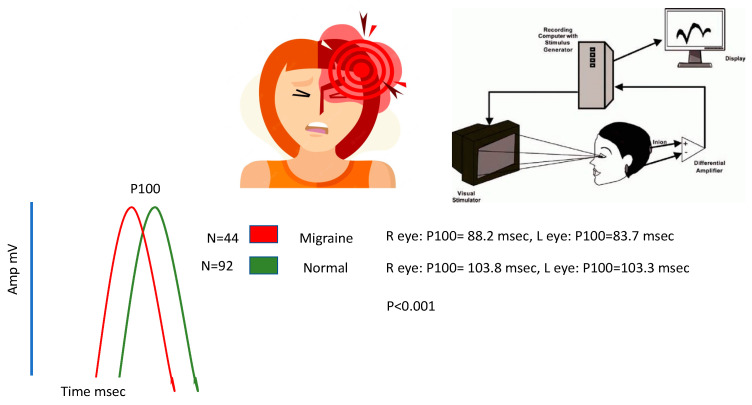
Representation of Migraine and VEP.

**Figure 3 biomedicines-11-02979-f003:**
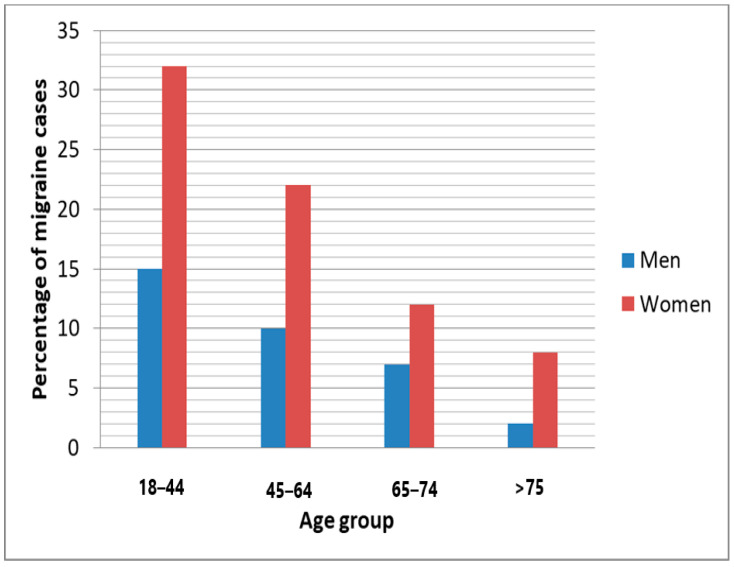
Age group-based percentage of migraine cases.

**Figure 4 biomedicines-11-02979-f004:**
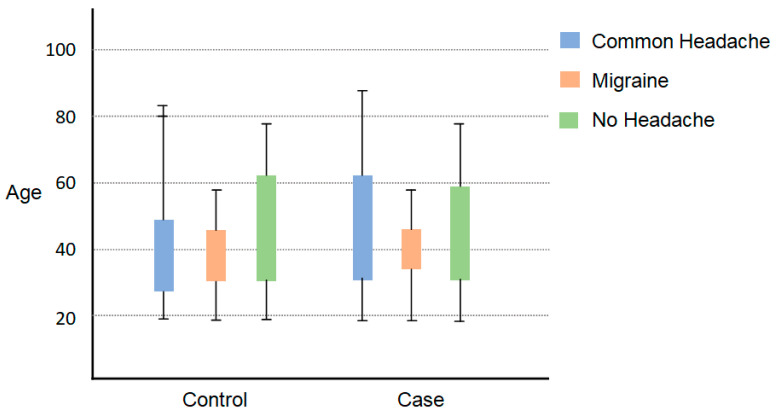
Comparison of P100 latency between both groups for the right eye.

**Figure 5 biomedicines-11-02979-f005:**
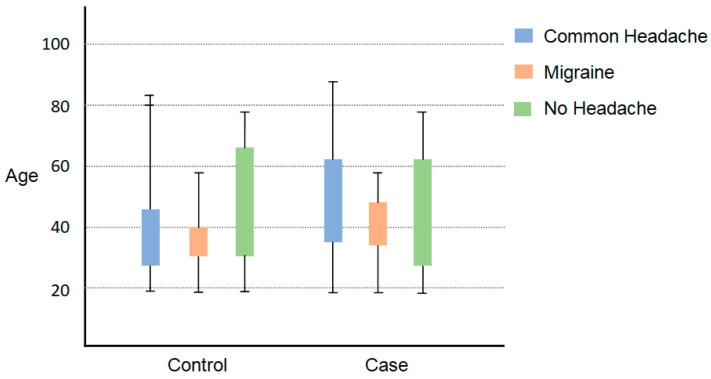
Comparison of P100 latency between both groups for the left eye.

**Table 1 biomedicines-11-02979-t001:** Gender and age distribution between migrainous and control (n = 136).

		Migraine n = 44	Control n = 92	*p*-Value
Gender	Male	15 (34.1%)	30 (33.3%)	0.8 *
	Female	29 (65.9%)	62 (67.7%)	
Age	Mean (±SD)	37.02 (±13.6)	34.1 (±12.3)	0.22 **

** Independent-samples *t*-test, * Chi-square test for association.

**Table 2 biomedicines-11-02979-t002:** Comparison between migraine and control.

	Migraine n = 44	Control n = 92	*p*-Values *
Left Eye			
N75	59.5 (±11.6)	74.6 (±9.1)	<0.0001
P100	83.7 (±7.2)	103.3 (±8.1)	<0.0001
N145	115.1 (±12.9)	136.8 (±14)	<0.0001
Amplitude	6.4 (±4.3)	8.8 (±5)	0.008
Right Eye			
N75	64.1 (±12.8)	75.4 (±11.4)	<0.0001
P100	88.2 (±13.5)	103.8 (±10.1)	<0.0001
N145	120.1 (±18.4)	137.8 (±16)	<0.0001
Amplitude	5.9 (±4)	8.9 (±4.5)	<0.0001

* By Independent-samples *t*-test.

**Table 3 biomedicines-11-02979-t003:** Logistic regression analysis regarding likelihood of migraine occurrence.

	B	S.E.	*p*-Values	Odds Ratios	95% C.I. for OR	
					Lower	Upper
P100 Left eye	0.925	0.297	0.002	2.521	1.409	4.51
P100 Right eye	0.073	0.049	0.137	1.076	0.977	1.185

**Table 4 biomedicines-11-02979-t004:** The relationship between left and right eye measures in both sets of subjects.

Group	Measurements	Correlation (r)	*p*-Values
Migraine	N75 Left Eye and N75 Right Eye	0.751	<0.0001
	P100 Left Eye and P100 Right Eye	0.394	<0.0001
	N145 Left Eye and N145 Right Eye	0.480	<0.0001
	Amplitude Left Eye and Amplitude Right Eye	0.739	<0.0001
Control	N75 Left Eye and N75 Right Eye	0.640	<0.0001
	P100 Left Eye and P100 Right Eye	0.774	<0.0001
	N145 Left Eye and N145 Right Eye	0.706	<0.0001
	Amplitude Left Eye and Amplitude Right Eye	0.842	<0.0001

By independent-samples *t*-test.

## Data Availability

Data are contained within the article.

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
