# Peer review of "Prospective Matched Case-Control Study of Over-Early P100 Wave Latency in Migraine with Aura"

_biomedicines, 2023, doi:10.3390/biomedicines11112979_

Round 1
Reviewer 1 Report (Previous Reviewer 1)
Comments and Suggestions for Authors
This paper by Alshamrani explores the relationship between visual evoked potentials (VEPs) and migraine with aura, a subtype of migraine characterized by visual disturbances. The study involved 136 participants, with 44 diagnosed with migraine with aura and 92 healthy controls. VEPs, which measure brain responses to visual stimuli, were recorded and analyzed. The results indicated that individuals with migraine with aura had altered VEP patterns compared to the control group. Additionally, the study examined factors like gender and age but found no significant correlations. This research provides insights into the potential use of VEPs as a diagnostic tool and suggests that neurological factors may play a role in migraine with aura.
Overall, I find the objective presented in this article to be quite intriguing, and the authors' insightful observations on this relevant subject matter could capture the attention of Biomedicines readers. However, there are certain points worth addressing, including specific comments and essential evidence required to bolster the author's argument. These adjustments are necessary to enhance the manuscript's quality, suitability, and overall readability before it can be published in its current state. In conclusion, I recommend publication of this research article once the author has thoughtfully incorporated the feedback and suggestions outlined below.
- The introduction provides a general overview of migraines but lacks a clear research objective. It's essential to explicitly state the research questions or hypotheses you aim to address in the paper.
- Ensure that you have included all relevant and up-to-date references, especially for recent findings in migraine research. It's important to demonstrate a comprehensive review of the literature.
- In the "Materials and Methods" section, authors describe the sample size and sampling method but do not mention the inclusion and exclusion criteria in detail. Providing more information about the criteria for participant selection will help readers assess the validity of the study.
- The section discussing bias and confounders is somewhat brief. It's crucial to explicitly describe how you addressed or controlled for potential sources of bias and confounding in your study. Furthermore, I suggest discussing how you handled variables such as age, gender, and other potential confounders in your statistical analysis.
The authors have diligently addressed the comments and feedback provided. They have made significant improvements to their manuscript in response to the suggestions. Therefore, I recommend only minor revisions at this stage, as the authors have effectively incorporated the feedback to enhance the quality of the paper.
Comments on the Quality of English Language
Minor English editing is required.
Author Response
Please see the attachment.

Reviewer 2 Report (Previous Reviewer 2)
Comments and Suggestions for Authors
Conclusion should be concise.
Study limitation and future direction should be discussed in Discussion section.
Round 2
Reviewer 2 Report (Previous Reviewer 2)
Comments and Suggestions for Authors
Now this manuscript seems to be acceptable.
This manuscript is a resubmission of an earlier submission. The following is a list of the peer review reports and author responses from that submission.
Round 1
Reviewer 1 Report
Comments and Suggestions for Authors
Alshamrani and colleagues in the present article entitled ‘Migraine with Aura linked to delayed P100 wave response in a case-control study', investigated how visual migraine could serve as a prototype abnormality of over-early P100, addressing how patients with migraine with aura have altered early visual processing that could potentially be related to pathophysiology of migraine.
In general, I think the idea of this article is really interesting and the authors’ fascinating observations on this timely topic may be of interest to the readers of Biomedicines. However, some comments, as well as some crucial evidence that should be included to support the author’s argumentation, needed to be addressed to improve the quality of the manuscript, its adequacy, and its readability prior to the publication in the present form. My overall judgment is to publish this paper after the authors have carefully considered my suggestions below, in particular reshaping parts of the ‘Introduction’ and ‘Methods’ sections by adding more evidence.
Please consider the following comments:
· Abstract: In my opinion, Authors should consider rephrasing this section. According to the Journal’s guidelines, the Abstract should contain most of the following kinds of information in brief form. Please, consider giving a more synthetic overview of the paper's key points: I would suggest rephrasing the results and conclusion to make them clear for readers to understand. Also, in my opinion, it would be better not to use abbreviations in this section.
· A graphical abstract that will visually summarize the main findings of the manuscript is highly recommended.
· Introduction: I suggest the authors to reorganize the Introduction section, which seems inhomogeneous and dispersive. I think that more information about pathomechanisms underlying migraine condition. Thus, I suggest the authors to make an effort to provide a brief overview of the pertinent published literature that offer a perspective on neurobiological alterations that lead to this neurological condition, because as it stands, this information is not highlighted in the text. Authors should briefly describe the involvement of neurogenic inflammation and neuropeptides in the pathophysiology of migraine, as well as deepen information on how electrophysiological tests may hold the key for a better understanding of migraine pathogenesis, providing evidence that focused on how altered brain rhythmic phenotypes are associated with specific neurological and neuropsychiatric diseases ((https://doi.org/10.3390/biomedicines10123189; https://doi.org/10.3390/biomedicines10081897).
· Bias, Confounders, and Statistical Analysis: I suggest rewriting this section more accurately.
· Discussion: In this final section, authors described the results of their study and their argumentation and captured the state of the art well; however, I would have liked to see some views on a way forward. I believe that the authors should make an effort, trying to explain the theoretical implication as well as the translational application of this paper, to adequately convey what they believe is the take-home message of their study. In this regard, I believe that it would be necessary to discuss theoretical and methodological avenues in need of refinement, as well as suggestions of a path forward in understanding.
· In my opinion, I think the ‘Conclusions’ paragraph would benefit from some thoughtful as well as in-depth considerations by the authors, because as it stands, it is very descriptive but not enough theoretical as a discussion should be. Authors should make an effort, trying to explain the theoretical implication as well as the translational application of their research.
· References: Authors should consider revising the bibliography, as there are several incorrect citations. Indeed, according to the Journal’s guidelines, they should provide the abbreviated journal name in italics, the year of publication in bold, the volume number in italics for all the references.
I hope that, after these careful revisions, this paper can meet the Journal’s high standards for publication.
I am available for a new round of revision of this article.
I declare no conflict of interest regarding this manuscript.
Best regards,
Reviewer
Reviewer 2 Report
Comments and Suggestions for Authors
Title indicated delayed response of p100 in migraine patients.
Howver, the results indicated shorted P100 latency.
This should be fully explained.
Author Response
Reviewer 2
1-Title indicated delayed response of p100 in migraine patients.
Howver, the results indicated shorted P100 latency.
This should be fully explained.
Reply: As per reviewer’s advice this has now been corrected. Everywhere throughout the manuscript it is now P100 latency.
Reviewer 3 Report
Comments and Suggestions for Authors
The paper is not suitable for publication. This is a resubmission. The authors have not followed previous suggestions from the careful revision process. The interpretation of results is tendentious. The overall quality is poor, as well as the whole design of the study... The study sample is low. The risk of bias is too high. The interpretation of VEPs is not in line with previous studies suggested in the review steps 1 and 2. The majority of references come from Coppola et al, but there is no mention of hyperexcitability, SIFI, and recent advances in the pathophysiology of migraine. I cannot approve the publication of this paper.
Round 2
Reviewer 2 Report
Comments and Suggestions for Authors
Conclusion should be final comments from this study.
Visual informations are processed in entire visual systems.
This study included only the patients with migraine aura, so this finding is not applicatble for the patients with migraine witout aura.
Reviewer 3 Report
Comments and Suggestions for Authors
There is no significant amelioration since the previous comments, despite being the third version of the manuscript. I do not want to repeat this again, the same pitfalls are still present.
Author Response
Dear Editor,
I hope this email finds you well. I am reaching out to inquire if there are any remaining comments or feedback that require my attention. I would appreciate it if you could kindly inform me of any outstanding edits or suggestions that I need to address.
Your valuable insights and recommendations play a crucial role in enhancing the quality of the work, and I want to ensure that I address any remaining concerns before finalizing the document. Your timely response would be greatly appreciated as it will allow me to make the necessary edits promptly.
Thank you for your attention and assistance throughout this process. I look forward to your reply.
Kind regards,
Please see the attachment.

Round 3
Reviewer 2 Report
Comments and Suggestions for Authors
Unfortunately author could not understand and respond my previous comments.
Reviewer 3 Report
Comments and Suggestions for Authors
Suggestions have not been followed yet.